# Impact of a 12-Week Dietary Intervention on Adipose Tissue Metabolic Markers in Overweight Women of Reproductive Age

**DOI:** 10.3390/ijms25158512

**Published:** 2024-08-04

**Authors:** Gita Erta, Gita Gersone, Antra Jurka, Peteris Tretjakovs

**Affiliations:** Department of Human Physiology and Biochemistry, Riga Stradins University, LV-1007 Riga, Latvia

**Keywords:** salivary amylase activity, overweight, diet intervention, adipose tissue, metabolic markers, GLP-1

## Abstract

The prevalence of overweight and obesity in women of reproductive age leads to significant health risks, including adverse metabolic and reproductive outcomes. Effective dietary interventions are critical to improving health outcomes in this population. This study investigates the impact of a 12-week diet intervention on metabolic markers of adipose tissue in overweight women of reproductive age, determining whether calorie restriction or low-starch diets are more effective, while also accounting for salivary amylase activity. A total of 67 overweight women of reproductive age were enrolled in a randomized controlled trial (RCT). Participants were divided into high-salivary-amylase (HSA) and low-salivary-amylase (LSA) groups based on baseline salivary amylase activity measured using a spectrophotometric method. Each group was further subdivided into two dietary intervention groups: calorie restriction (CR) and low starch (LS), resulting in four subgroups (HSA-CR, HSA-LS, LSA-CR, LSA-LS), along with a control group (CTR) of normal-weight individuals (no intervention). Participants were assigned to a calorie-restricted diet or a low-starch diet for 12 weeks. Key metabolic markers of adipose tissue, including insulin sensitivity, adipokines, cytokines, and lipid profiles, were measured at baseline (T0), 30 min after consuming starch-containing muesli (T1), and 12 weeks after intervention (T2). Active GLP-1, glucagon, and C-peptide levels were assessed to clarify the hormonal mechanisms underlying the dietary effects. Salivary amylase activity was also measured to examine its role in modulating glucose and GLP-1 responses. Both diet interventions led to significant improvements in metabolic markers of adipose tissue, though different ones. Calorie restriction improved insulin sensitivity by effectively reducing visceral fat mass and enhancing insulin signaling pathways. In contrast, the low-starch diet was linked to a reduction in the coefficient of glucose variation influenced partly by changes in GLP-1 levels. Our findings highlight the importance of personalized diet strategies to optimize metabolic health in this demographic.

## 1. Introduction

Adipose tissue plays a crucial role in regulating metabolic homeostasis, functioning not only as a site for energy storage but also as an active endocrine organ. It secretes various adipokines and cytokines that influence systemic metabolism and insulin sensitivity [1,2,3]. In the context of overweight and obesity, adipose tissue undergoes functional changes that can lead to metabolic dysfunctions, including insulin resistance, chronic inflammation, and dyslipidemia [4]. These changes are particularly critical for women of reproductive age, as they impact not only metabolic health but also reproductive outcomes [5,6,7].

Calorie restriction and adjustments in macronutrient composition, such as low-starch diets, have shown promising results in improving metabolic markers [8]. Calorie restriction, characterized by a reduction in total caloric intake without inducing nutrient deficiencies, has been linked to enhanced insulin sensitivity, improved lipid profiles, and reduced inflammatory markers [9,10]. Similarly, low-starch diets, which limit the intake of rapidly digestible carbohydrates, have been shown to positively influence glycemic control and decrease adiposity [11].

Dietary interventions impose their effects primarily through the modulation of gut hormones, such as glucagon-like peptide-1 (GLP-1). GLP-1, an incretin hormone, is essential for glucose homeostasis by enhancing insulin secretion and inhibiting post-prandial glucagon release [12,13]. The secretion of GLP-1 is influenced by dietary composition and the rate of digestion and absorption of carbohydrates [14].

Salivary amylase, an enzyme that initiates starch hydrolysis in the oral cavity, has been identified as a significant contributor to this process [15]. Variability in salivary amylase activity among individuals can influence post-prandial glucose levels and subsequent GLP-1 secretion [16,17]. Elevated salivary amylase activity facilitates rapid starch degradation and glucose absorption, potentially increasing GLP-1 release and enhancing metabolic outcomes. Conversely, reduced salivary amylase activity results in slower starch digestion and altered glycemic responses [18,19] which may affect GLP-1 dynamics and adipose tissue metabolism.

Despite the growing body of evidence on the role of dietary interventions and salivary amylase activity in metabolic health, comprehensive studies are needed that focus on women of reproductive age, a demographic particularly vulnerable to the adverse effects of excess adiposity. Understanding how different diet patterns influence metabolic markers of adipose tissue and the underlying physiological mechanisms in this population is essential for developing targeted nutritional strategies.

This study aims to investigate the impact of a 12-week dietary intervention, comprising calorie restriction and low-starch intake, on adipose tissue metabolic markers in overweight reproductive-age women. We hypothesize that this intervention will improve key metabolic markers, potentially mediated by changes in GLP-1 levels and modulated by individual variations in salivary amylase activity. By clarifying these relationships, our research seeks to contribute to the development of personalized dietary recommendations that can improve metabolic health and reproductive outcomes in this population.

## 2. Results

In the LSA-LS participant group, the Wilcoxon signed rank test was used to compare active GLP-1 levels at baseline and post-intervention. The analysis revealed a statistically significant increase in active GLP-1 levels, with median values increasing from 12.65 at baseline to 60.38 after the low-starch diet (W = 45, *p* = 0.001) (Figure 1).

The Kruskal–Wallis H test demonstrated significant differences in glucagon levels between the dietary intervention groups (H = 10.97, *p* = 0.0035). Subsequent post hoc pairwise comparisons indicated that glucagon levels were significantly lower in the compared groups (Figure 2).

These findings suggest that a combination of a low-amylase diet and calorie restriction is associated with a substantial reduction in glucagon levels, which may benefit the management of glucagon levels and improve fasting glucose levels.

Descriptive statistics indicated that the median glucagon level in the LSA-CR group before the intervention was 19 pg/mL (IQR: 9.107–63.90 pg/mL), which decreased to a median of 13.13 pg/mL (IQR: 11.3–25.71 pg/mL) after the intervention.

Additionally, the Kruskal–Wallis H test identified significant differences in triglyceride levels among the five groups at baseline, before the intervention (H = 11.22, *p* = 0.0158) (Figure 3), indicating initial variability in the metabolic profiles of the participants. However, post-intervention, these differences were no longer significant, suggesting that the dietary interventions normalized triglyceride levels across groups, effectively reducing the initial disparities. This normalization implies that both calorie-restricted and low-starch diets were equally effective in modulating triglyceride levels, irrespective of baseline differences.

Furthermore, the Kruskal–Wallis H test revealed significant differences in C-peptide levels among the diet intervention groups (H = 5.0, *p* = 0.0222) (Figure 4) and in leptin levels (H = 38.42, *p* = 0.0004) (Figure 5). These significant differences underscore the impact of dietary interventions on these biomarkers.

We measured Spearman’s rank coefficient between salivary amylase activity and various adipose tissue markers in 67 women at three time points: baseline (T0), 30 minutes after consuming starch-containing muesli (T1), and 12 weeks after the diet intervention (T2) (Figure 6).

The analysis highlights the relationships between enzyme activity and adipose tissue markers at different points in time, providing insights into the effects of dietary intervention. While most parameters were measured at all three time points, glucose, C-peptide, calculated parameters (HOMA2IR, HOMA2-%S, and HOMA2-%B), and triglycerides were measured at the start and after 12 weeks.

Measuring these biomarkers provides insight into how dietary interventions affect various aspects of glucose homeostasis, offering critical insights for developing effective nutritional strategies to manage and prevent metabolic diseases (Table 1 and Table 2)

## 3. Discussion

In this study, we investigated the effects of different dietary interventions (low-starch diet and calorie restriction diet) on GLP-1, glucagon, C-peptide, and various metabolic markers in overweight women of reproductive age. Participants were categorized according to their salivary amylase activity (low and high) and compared to a control group with normal weight. Our findings provide new insights into how dietary modifications and individual enzymatic differences impact key metabolic biomarkers.

### 3.1. Glucagon

Role in Glucose Homeostasis: Glucagon, a hormone produced by pancreatic alpha cells, counterbalances insulin effects by stimulating hepatic glucose production through glycogenolysis and gluconeogenesis, raising blood glucose levels during fasting states or hypoglycemia [20].Relevance in Dietary Studies: Monitoring glucagon levels elucidates how different diets impact hepatic glucose production and overall glucose homeostasis. Diets that modulate glucagon secretion could influence fasting glucose levels and glucose variability, crucial components of metabolic health [21].

Our analysis showed significant differences in glucagon levels between the groups. Participants on a calorie restriction diet exhibited significantly lower glucagon levels after 12 weeks compared to those on a low-starch diet (H = 10.97, *p* = 0.0035) (Figure 2). This reduction in glucagon could contribute to improved glycemic control, as glucagon is known to raise blood glucose levels by stimulating hepatic glucose production [21,22].

### 3.2. C-Peptide

Role in Glucose Homeostasis: C-peptide, co-secreted with insulin from pancreatic beta cells in equimolar amounts, serves as a marker of endogenous insulin production [23]. Unlike insulin, the C-peptide is not extracted by the liver, making it a reliable measure of beta-cell function.Relevance in Dietary Studies: Assessing C-peptide levels helps determine how dietary interventions affect insulin secretion and beta-cell function. This information is essential for understanding the long-term impacts of diets on insulin production and the risk of developing insulin resistance or type 2 diabetes.

C-peptide levels varied significantly between the groups (Figure 4). The control group with normal weight had significantly lower C-peptide levels compared to the overweight groups, both at fasting and post-prandially. Within the intervention groups, those on a calorie restriction diet with high salivary amylase activity had a more pronounced reduction in C-peptide levels over time.

### 3.3. GLP-1

Role in Glucose Homeostasis: Glucagon-like peptide-1 (GLP-1) is an incretin hormone secreted by enteroendocrine L-cells in the intestine in response to food intake. It enhances glucose-dependent insulin secretion, inhibits glucagon release, and slows gastric emptying, collectively contributing to post-prandial glucose control [24,25].Relevance in Dietary Studies: Diet-induced changes in GLP-1 levels significantly affect post-prandial glucose metabolism. Measuring GLP-1 provides insights into how different dietary patterns influence incretin response, satiety, and insulin sensitivity. This is particularly relevant for developing dietary strategies to improve glucose regulation and manage diabetes.

The Kruskal–Wallis test revealed significant differences in GLP-1 levels among the different intervention groups at various time points (H = 10.97, *p* = 0.0035) (Figure 1). Notably, post-dietary intervention analysis (T2) indicated that GLP-1 levels increased significantly more in the low-starch diet group with low salivary amylase activity compared to the other groups. This suggests that individuals with lower salivary amylase activity may experience increased GLP-1 secretion in response to starch intake, potentially enhancing satiety and glucose homeostasis (Figure 1).

### 3.4. Leptin Levels

Leptin, a hormone produced primarily by adipocytes, regulates energy balance and body weight. Our study observed a significant decrease in leptin levels in the CR groups compared to the baseline and control groups (H = 38.42, *p* = 0.0004) (Figure 5). This is consistent with previous research that indicates that caloric restriction reduces leptin levels due to decreased fat mass and reduced adipocyte size [26]. However, the LS groups did not show a significant reduction in leptin levels, suggesting that macronutrient composition could play a lesser role than caloric intake in modulating leptin.

### 3.5. GDF-15 Levels

Growth differentiation factor 15 (GDF-15) is a stress-induced cytokine linked to metabolic regulation [27,28,29]. Our findings did not show a significant change. Overall, while our dietary interventions were successful in addressing some metabolic markers, the lack of significant differences in GDF-15 changes suggests that this cytokine’s regulation is multifaceted and may not be easily modified by diet alone in the short term (Figure 7).

### 3.6. Implications and Future Directions

Long-term Effects and Sustainability: Future studies should investigate the long-term effects and sustainability of different diet interventions. This could include reviewing the maintenance of improvements of metabolic health and adherence to dietary regimens for extended periods.Personalized Nutrition Approaches: Given the variability in individual responses based on salivary amylase activity, future research should focus on developing personalized nutrition strategies. This includes exploring how genetic and phenotypic differences can guide personalized diet recommendations to improve metabolic health.Mechanistic Studies: More mechanistic studies are needed to elucidate the underlying pathways through which diet interventions affect metabolic markers. This involves exploring the molecular and cellular mechanisms in adipose tissue that respond to different dietary components.

Collectively, these findings underscore the importance of personalized dietary interventions in managing metabolic health. Future research should continue to explore these relationships to develop more effective and individualized dietary strategies for the prevention and management of metabolic diseases.

Elucidating the role of glucagon in different dietary contexts can open new avenues to improve glycemic control in overweight individuals. Future research should examine the long-term effects of these diet interventions and explore changes in the expression of genes involved in the lipid metabolism, inflammation, and insulin signaling pathways due to dietary interventions. Techniques such as quantitative PCR and RNA sequencing are essential for identifying these alterations in gene expression.

Additionally, studies with larger and more diverse cohorts are necessary to validate these findings and determine their generalizability to broader populations. By tailoring dietary interventions to individual metabolic profiles, we can enhance the efficacy of treatments aimed at mitigating obesity-related metabolic disorders in reproductive-age women.

### 3.7. Limitations

The duration of the study was limited to 12 weeks, but larger, longer-term studies are required to confirm our findings. Additionally, we did not control other lifestyle factors, such as physical activity, that could influence metabolic outcomes. Despite these limitations, our findings provide valuable insights into the role of diet and enzymatic activity in metabolic regulation.

### 3.8. Key Findings

Improvements in Metabolic Markers: Both calorie-restricted (CR) and low-starch (LS) diets led to significant improvements in metabolic markers, although of different kinds.Impact of Salivary Amylase Activity: Participants with a higher baseline salivary amylase activity (HSA) showed better responses to the CR diet in terms of insulin sensitivity, although this finding should be interpreted with caution due to variability in individual responses.

Individuals with low salivary amylase (LSA) activity exhibited superior glycemic control on low-starch diets. However, further research is needed to confirm these results and understand the underlying mechanisms.

### 3.9. Personalized Nutrition

The study underscores the importance of tailoring dietary interventions based on individual metabolic profiles, specifically salivary amylase activity, to optimize metabolic outcomes.

### 3.10. Detailed Findings

Leptin Levels: Significant reductions in leptin levels were observed in the LSA-CR group from T0 to T2, indicating the effectiveness of calorie restriction diets in decreasing leptin levels in participants with low salivary amylase activity. Interestingly, a more pronounced increase in leptin levels was observed after consuming starch-containing muesli (T1) in both LSA groups compared to those with high salivary amylase activity.

It can be attributed to slower starch digestion and absorption of starch in LSA individuals [15]. This results in prolonged elevation of blood sugar and a sustained insulin response, which may influence further activation of the PI3K/Akt pathway [29], thereby enhancing the transcription and translation of leptin in adipocytes. Studies have shown that insulin can upregulate leptin mRNA expression and increase leptin secretion in adipose tissue [11].

### 3.11. Insulin Sensitivity (HOMA2-%S)

The LSA-CR group exhibited significant improvements in the HOMA2-%S score, reflecting enhanced insulin sensitivity (Figure 8).

Physiological Mechanism: Calorie restriction can result in a more effectively reduced fat mass and enhanced insulin signaling pathways.Overall Metabolic Improvements: The 12-week dietary intervention resulted in significant metabolic improvements in overweight women of reproductive age, highlighting the importance of considering salivary amylase activity.

### 3.12. Mechanistic Insights

The observed changes in metabolic markers can be attributed to several mechanisms.

Caloric Restriction: It likely reduces adipocyte size and fat mass, leading to lower leptin production and enhanced insulin sensitivity [26].Low-starch Diet: Reduces post-prandial glucose spikes and reduces hyperinsulinemia, which collectively contribute to better metabolic health [30,31]

### 3.13. Personalized Responses Based on Salivary Amylase Activity

Our study also highlights the importance of personalized nutrition, particularly the role of salivary amylase activity in modulating metabolic responses to diet interventions. There is the potential to use salivary amylase activity as a biomarker for personalized dietary recommendations.

### 3.14. Implications and Future Directions

Personalized Nutrition Approaches: The findings highlight the potential of personalized nutrition strategies based on salivary amylase activity to enhance the efficacy of dietary interventions for metabolic health.Long-term Effects and Sustainability: Future studies should investigate the long-term effects and sustainability of different diet interventions, including gene expression profiling to understand changes in lipid metabolism, inflammation, and insulin signaling pathways.Larger, Diverse Cohorts: Studies with larger and more diverse cohorts are necessary to confirm these findings and determine their generalizability to broader populations.

## 4. Materials and Methods

### 4.1. Study Design and Participants

This study was conducted with 67 women of reproductive age (18–45 years) with a BMI between 25 and 29.9 kg/m^2^ kg/m^2^ (Table 3). Participants were recruited from a health center and screened for eligibility. Exclusion criteria include pregnancy, lactation, chronic diseases (e.g., diabetes, cardiovascular disease), and the use of medications that affect metabolism. The study protocol was approved by the Ethics Committee of Rigas Stradiņš University (Ethics Committee number: 22-2/479/2021), and all participants provided their written informed consent in accordance with the Declaration of Helsinki.

### 4.2. Dietary Interventions

Participants were randomly assigned to one of two dietary intervention groups for a duration of 12 weeks.

Low-starch Diet Group: This group followed a low-starch diet, emphasizing the consumption of low-glycemic-index vegetables, proteins, and healthy fats. Daily starch intake was limited to less than 50 g.Caloric Restriction Group: This group followed a caloric restriction diet, reducing their daily caloric intake by 500 kcal from their estimated energy requirement, calculated based on the Harris–Benedict equation.

Dietary adherence was monitored through weekly food diary and online consultations with an endocrinologist.

### 4.3. Evaluation of Salivary Amylase Activity

Salivary amylase activity was evaluated using the Salimetrics Amylase Activity Assay (Salimetrics, State College, PA, USA). Unstimulated saliva samples were collected in the morning after an overnight fast to ensure consistency. Salivary amylase activity was determined according to the manufacturer’s protocol. Participants were classified into high- and low-salivary-amylase-activity groups based on the median split of amylase activity data.

### 4.4. Biochemical and Molecular Analysis

Fasting blood samples were collected at baseline (T0), after the consumption of starch-containing muesli (T1), and after the 12-week dietary intervention (T2). The following metabolic markers were analyzed:Active GLP-1: Measured using a GLP-1 (Active) ELISA kit (Millipore, Billerica, MA, USA).C-peptide: Assessed using a chemiluminescent immunoassay kit (IMMULITE 2000, Siemens Healthineers, Erlangen, Germany).Glucagon: Measured using the Glucagon ELISA Kit (Cat. No. EZGLU-30K, Millipore Sigma, Burlington, MA, USA).HOMA2-%S: Calculated using fasting insulin and glucose levels with the HOMA calculator (version 2.2.3, Diabetes Trials Unit, University of Oxford).Leptin: Measured using a multiplex immunoassay (Luminex, Austin, TX, USA).GDF-15: Quantified using ELISA kits (R&D Systems, Minneapolis, MN, USA).Triglycerides: Measured using an enzymatic colorimetric method with a commercial kit (Roche Diagnostics, Basel, Switzerland).Visceral fat: Measured using a bioimpedance scale (Omron BF511, Omron Healthcare, Kyoto, Japan).

### 4.5. Study Groups and Interventions

Participants were divided into two groups according to their baseline salivary amylase activity: High salivary amylase (HSA) and low salivary amylase (LSA). Each salivary amylase group was further subdivided into two dietary intervention groups.

Calorie Restriction (CR) Group: Participants followed a calorie-restricted diet, reducing their daily caloric intake by 500 kcal.Low-Starch (LS) Diet Group: Participants followed a low-starch diet, limiting their daily intake of starch to less than 50 g.

This resulted in four subgroups: HSA-CR, HSA-LS, LSA-CR, LSA-LS. Additionally, a control group (CTR) of normal weight was included (Table 4).

### 4.6. Dietary Intervention

Dietary interventions were designed by registered dietitians and included meal plans and educational sessions to ensure adherence. Participants received weekly online counseling and were required to maintain a food diary. Compliance was monitored through regular check-ins and food diary reviews.

### 4.7. Sample Collection and Analysis

Saliva Samples: Collected in the morning after an overnight fast. Salivary amylase activity was measured using an enzymatic assay kit (Salimetrics, State College, PA, USA) according to the manufacturer’s instructions.Blood Samples: Fasting blood samples were analyzed for metabolic markers as described above.

### 4.8. Ethical Approval

The study was conducted in accordance with the Declaration of Helsinki and was approved by the Ethics Committee of Rigas Stradiņš University (Ethics Committee number: 22-2/479/2021). Written informed consent was obtained from all participants prior to their enrollment in the study.

This comprehensive methodology ensures a rigorous assessment of the dietary interventions’ impact on metabolic health markers in the target population.

### 4.9. Statistical Analysis

#### Data Analysis

Statistical analyses were conducted using GraphPad Prism 10. Due to the non-normal distribution of the data, non-parametric tests were employed. Descriptive statistics were summarized as medians with interquartile ranges (IQRs) for each variable.

Descriptive Statistics: Median and IQR were calculated for each variable (e.g., leptin, GDF-15, HOMA2-%B) at each time point (T0, T1, T2) across all groups (HSA-CR, HSA-LS, LSA-CR, LSA-LS, and control) (Table 1).

Spearman’s rank correlation coefficients were calculated to assess the relationships between salivary amylase activity and metabolic markers. Statistical significance was set at *p* < 0.05 (Figure 6).

Group Comparisons: The Kruskal–Wallis H test was used to assess differences between the five groups at each time point. For post hoc pairwise comparisons, Dunn’s multiple comparison test with Bonferroni correction was used to determine which specific groups differ from each other (Table 2)Within-group Comparisons: Changes within each group over time (T0 to T1, T1 to T2, T0 to T2) were analyzed using the Wilcoxon signed rank test.Effect of Dietary Interventions: To evaluate the impact of diet interventions on the primary outcomes, the Friedman test was used to compare values at T0, T1, and T2 within each subgroup, followed by post hoc analysis with Dunn’s test.

## 5. Conclusions

This study underscores the significant impact of dietary interventions on metabolic health markers in overweight women of reproductive age, with a particular focus on the roles of calorie restriction (CR) and low-starch (LS) diets. Both CR and LS diets led to notable improvements in metabolic markers, including reduced leptin levels and enhanced insulin sensitivity, as indicated by the HOMA2-%S scores.

A key finding of this study is the differential response to dietary interventions based on salivary amylase activity. Participants with higher baseline HSA exhibited superior improvements in insulin sensitivity when following the CR diet, while individuals with low salivary amylase activity (LSA) exhibited superior glycemic control on low-starch diets.

The study also suggests that variations in salivary amylase activity are associated with different compositions of the gut microbiota, which in turn influence metabolic health.

These findings emphasize the potential for the use of salivary amylase activity as a biomarker to tailor dietary recommendations, thus optimizing metabolic outcomes. Personalized diet interventions could enhance the effectiveness of treatments aimed at reducing obesity-related metabolic disorders in reproductive-age women.

Future research should focus on the long-term effects of these dietary interventions, including gene expression profiling, to understand changes in lipid metabolism, inflammation, and insulin signaling pathways. Larger and more diverse cohorts are needed to validate these findings and determine their generalizability. Additionally, further mechanistic studies are essential to elucidate the pathways through which dietary interventions affect metabolic markers.

Overall, this study provides valuable insights into the role of diet and enzymatic activity in metabolic regulation and underscores the importance of considering individual metabolic profiles when designing dietary interventions for metabolic health.

## Figures and Tables

**Figure 1 ijms-25-08512-f001:**
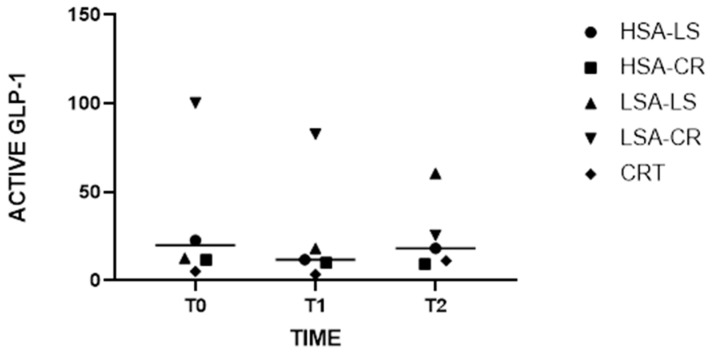
Medians of Active GLP-1 Levels at Baseline and Post-Intervention with Grand Median Across All Groups.

**Figure 2 ijms-25-08512-f002:**
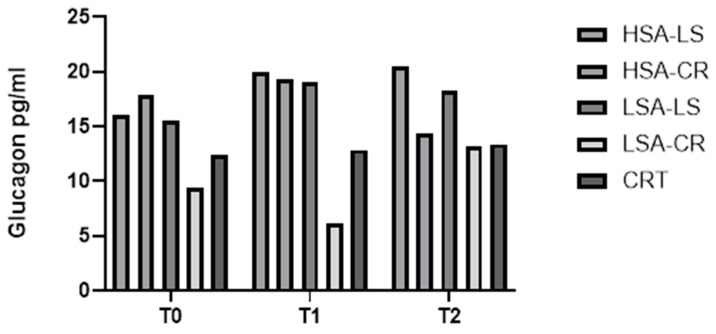
Glucagon.

**Figure 3 ijms-25-08512-f003:**
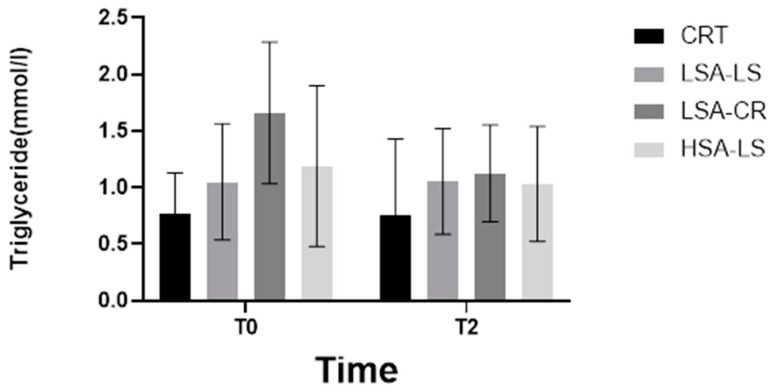
Triglyceride.

**Figure 4 ijms-25-08512-f004:**
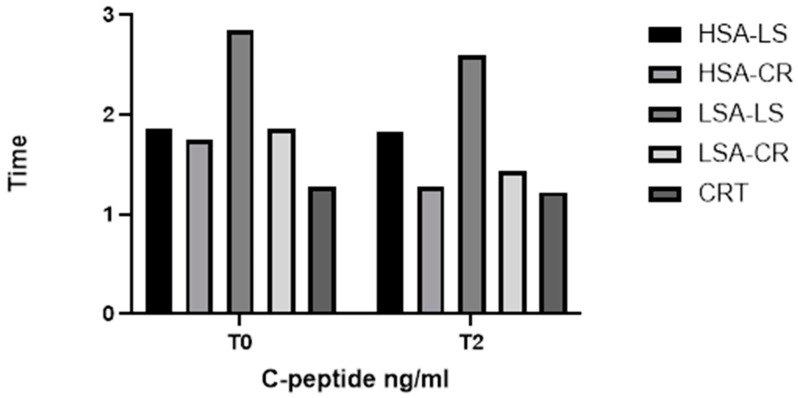
C-peptide.

**Figure 5 ijms-25-08512-f005:**
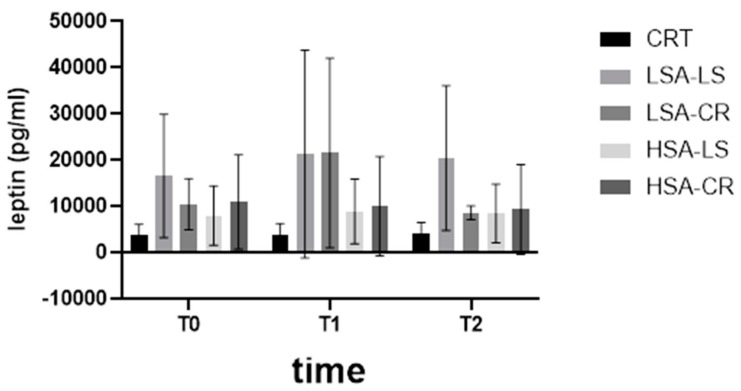
Leptin.

**Figure 6 ijms-25-08512-f006:**
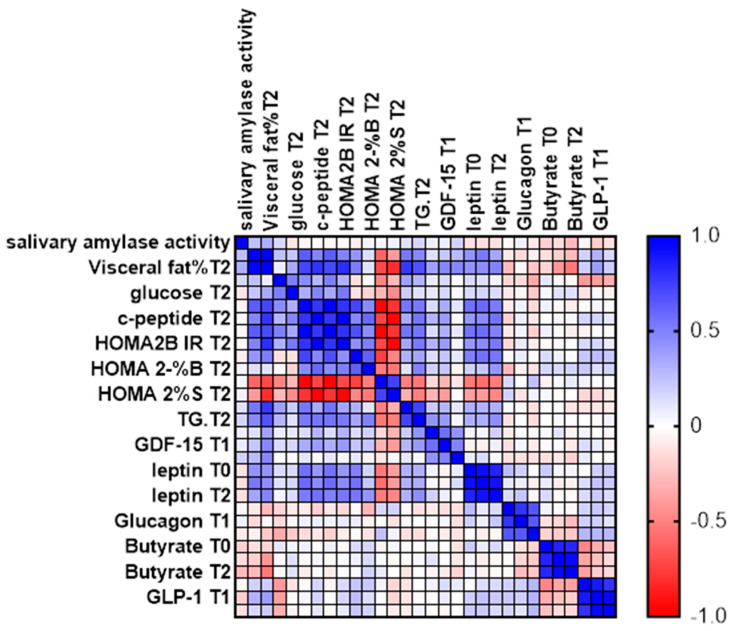
Spearman’s Rank Coefficient.

**Figure 7 ijms-25-08512-f007:**
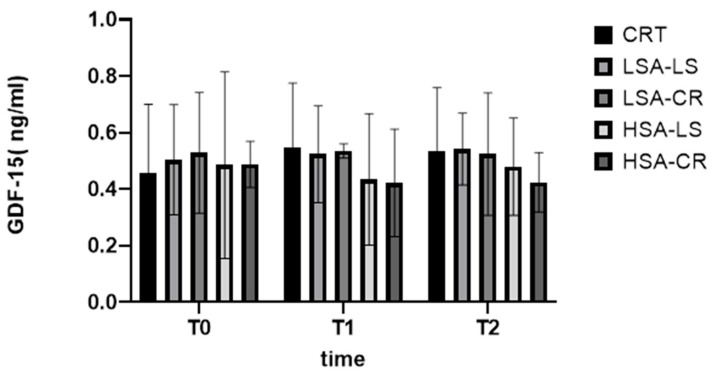
GDF-15.

**Figure 8 ijms-25-08512-f008:**
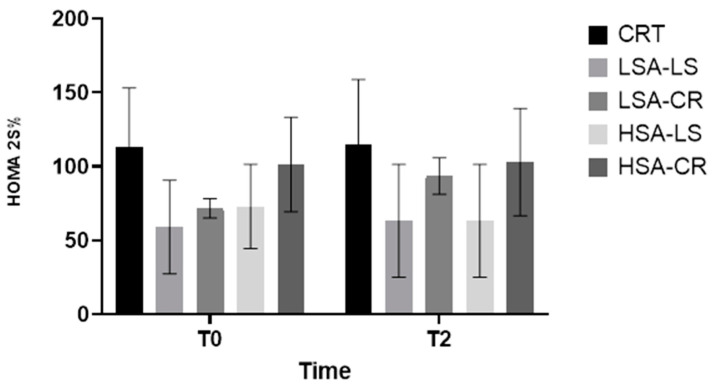
HOMA2-%S.

**Table 1 ijms-25-08512-t001:** Key statistical results. Descriptive statistics.

Biomarker	Group	Time Point	Median (IQR)
Active GLP-1	LSA-LS	T0	12.65 pg/mL (3.428–128.0)
Active GLP-1	LSA-LS	T2	60.38 pg/mL (7.93–224.3)
Glucagon	HSA-CR	T0	17.85 pg/mL (12.02–31.75)
Glucagon	HSA-CR	T2	14.40 pg/mL (8.33–34.28)
C-peptide	LSA-LS	T0	2.31 ng/mL (1.26–3.3)
C-peptide	LSA-LS	T2	2.1 ng/mL (0.9–2.6)
Leptin	HSA-CR	T0	7146 pg/mL (4879–13,123)
Leptin	HSA-CR	T2	5607 pg/mL (4452–10,536)

**Table 2 ijms-25-08512-t002:** Key statistical results. Group comparisons.

Statistic/Analysis	Method	Result
Group comparisons	Kruskal–Wallis H test	Differences in glucagon levels between groups (H = 10.97, *p* = 0.0035). Glucagon levels significantly lower in high-amylase calorie restriction group
	Kruskal–Wallis H test	Differences in C-peptide levels (H = 5.0, *p* = 0.0222)
	Kruskal–Wallis H test	Differences in leptin levels (H = 38.42, *p* = 0.0004)
Within-group Comparisons	Wilcoxon signed rank test	Significant increase in active GLP-1 levels in LSA-LS group from 12.65 to 60.38 (W = 45, *p* = 0.001)
Dietary interventions	Friedman test followed by Dunn’s test with Bonferroni correction	Post hoc analysis revealed a significant increase in active GLP-1 levels from T0 (median: 12.65) to T2 (median: 60.38), with a significant *p*-value (*p* = 0.001)

**Table 3 ijms-25-08512-t003:** Demographic Information.

Characteristic	
Age (years)	
Mean (SD)	29.5 ± 6.2
Range	18–45
BMI (kg/m^2^)	
Mean (SD)	27.8 ± 2.1
Range	25–29.9
Adherence (%)	
Mean (SD)	85 ± 10
Range	60–100

**Table 4 ijms-25-08512-t004:** Distribution of participants in control and study groups.

Group	Number of Participants
Control Group	7
HSA-CR	15
HSA-LS	15
LSA-CR	15
LSA-LS	15
Total	67

## Data Availability

Data are contained within the article.

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
