# Peer review of "Impact of a 12-Week Dietary Intervention on Adipose Tissue Metabolic Markers in Overweight Women of Reproductive Age"

_ijms, 2024, doi:10.3390/ijms25158512_

Round 1

Reviewer 1 Report

Comments and Suggestions for Authors

Impact of a 12-Week Dietary Intervention on Adipose Tissue Metabolic Markers in Overweight Women in Reproductive-Age

The current topic of this research is auspicious considering that obesity and metabolic  dysfunctions have reached pandemic level. This study underscores the significant impact of dietary interventions on metabolic health markers in overweight women of reproductive-age women, with a particular focus on the roles of calorie restriction (CR) and low-starch (LS) diets. Both CR and LS diets led to notable improvements in metabolic markers, including reduced leptin levels and enhanced insulin sensitivity, as indicated by the HOMA2S% scores. A key finding of this study is the differential response to dietary interventions based on salivary amylase activity. Participants with higher baseline HSA exhibited superior improvements in insulin sensitivity when following the CR diet, while individuals with low salivary amylase activity (LSA) exhibited superior glycemic control on low starch diets. The study also suggests that variations in salivary amylase activity are associated with different compositions of the gut microbiota, which in turn influence metabolic health. Research included five study groups, the obtained results being processed through appropriately chosen statistical tests. Their presentation was made in diagrams for all dosed parameters. Conclusions must not contain a table and be too many with irrelevant ideas. References must be up dated.

Author Response

 Dear Reviewer, 

Thank you for your thorough and insightful review of our manuscript. We appreciate your positive feedback and the valuable suggestions you have provided. Here is our response to your comments: 

1. Updated References: We acknowledge the importance of citing the most current and relevant literature. We will review our reference list and update it to include more recent studies that align with the scope and findings of our research.  

2. Key Findings: We are pleased that you recognized the significance of our key findings, particularly the differential response to dietary interventions based on salivary amylase activity  We will ensure that these findings are clearly emphasized in the revised manuscript. 

3. Statistical Analysis: We have carefully chosen appropriate statistical tests to process the results from our study groups. We will provide a more detailed explanation of these tests in the methods section to enhance transparency and reproducibility. 

We are committed to addressing these points comprehensively in our revised manuscript and believe that these changes will significantly improve the quality and clarity of our paper. Thank you once again for your valuable feedback. 

Sincerely, 

Dr. Gita Erta 

Reviewer 2 Report

Comments and Suggestions for Authors

Comments for authors:

The paper by Erta et al. presents a comprehensive study on the effects of a 12-week dietary intervention on adipose tissue metabolic markers in overweight women of reproductive age. The study is well-designed, with a clear hypothesis and robust methodology. However, there are several areas where clarity, detail, and scientific rigor could be improved. The main issue is that the key findings reported in section 3.8 are too overreaching and not fully supported by the results.

I have no major suggestions for improvement, however, I have some minor points which I think the authors should take into consideration:

Caption of figure 8 is confusing and needs to be expanded to explain what the figure is showing. Are these correlations in all the participants combined? It seems like there are two squares for each variable, is this intended?

In figures 1 to 7, indicate which bars were significantly lower than others with an asterisk, this would be helpful to readers. 

In lines 94-96, the authors should indicate which group they are referring to here. 

Looking at fig 3, we can see that triglyceride levels varied greatly between the different groups even at baseline. The authors should mention that this was the case before intervention. 

Authors should provide the demographic information or “characteristics” of the groups studied in a table which should be presented by intervention group, with statistical tests to show if there are any differences in characteristics such as age, weight, BMI, were these not measured? 

Did the authors measure adherence to diet or weight before and after intervention? It would be helpful if they could report on these or mention them in the limitations. 

In the discussion, the authors should compare their study to other studies of high and low salivary amylase. The suggestion of personalised nutrition based on salivary amylase is an unusual recommendation and the authors should highlight other studies that support this suggestion. 

Overall, the study is well-conceived and executed, with important findings that contribute to the understanding of dietary interventions in metabolic health. Addressing the above points would enhance the clarity, rigor and impact of the paper.

Author Response

Dear Reviewer, 

Thank you for your detailed and constructive feedback on our manuscript. We appreciate your recognition of the strengths of our study and your thoughtful suggestions for improvement. Below, we address each of your points: 

1Clarity of Key Findings in Section 3.8: We acknowledge that the key findings in section 3.8 may appear overreaching. We will revise this section to ensure that our conclusions are more closely aligned with the presented data, clearly delineating which results support our hypotheses and which suggest areas for further investigation. 

2.Caption of Figure 6 (previous 8): We will expand the caption of Figure 6 to provide a clearer explanation of what the figure represents. We will specify whether the correlations include all participants combined and clarify the meaning of the two squares for each variable. 

3.Clarification of Group Reference in Lines 94-96: We will revise lines 94-96 to clearly specify which group is being referred to, ensuring there is no ambiguity for the reader. 

4. Baseline Triglyceride Levels in Figure 3: We will add a statement acknowledging the variability in triglyceride levels between different groups at baseline. This will provide a better context for the interpretation of the intervention effects. 

5.Demographic Information and Characteristics Table: We will include a table(3) presenting the demographic characteristics of the participants 

By addressing these points, we aim to improve the clarity, rigor, and overall impact of our paper. We are committed to refining our manuscript to meet the high standards expected in scientific research. 

Thank you once again for your valuable feedback. 

Sincerely, 

 Dr. Gita Erta